# Sustainable Lightweight Concrete Made of Cement Kiln Dust and Liquefied Polystyrene Foam Improved with Other Waste Additives

**Abeer M. El-Sayed [1], Abeer A. Faheim [1], Aida A. Salman [1] and Hosam M. Saleh [2,*]**

1   Chemistry Department, Faculty of Science, Al Azhar University, Cairo 11651, Egypt
2   Radioisotope Department, Nuclear Research Center, Egyptian Atomic Energy Authority, Cairo 11787, Egypt
*   Correspondence: hosam.saleh@eaea.org.eg

**Abstract:** The main objective of this study is to mix two problematic wastes, cement kiln dust (CKD) and polystyrene waste liquified by gasoline, to produce a new lightweight cementitious material, as a green composite used in the construction industry. Various ratios of liquified polystyrene (LPS) were blended with CKD to achieve the optimum mixing ratio in the absence and presence of different additives. A significant improvement of mechanical properties (compressive strength of 2.57 MPa) and minimization of the porosity (51.3%) with reasonable water absorption (42.4%) has been detected in the mixing of 30% LPS with CKD due to filling the voids and gaps with liquified polymer. Portland cement, waste glass, and iron slag have been incorporated into CKD-30% LPS paste at different mass fractions of 0%, 5%, 10%, 15%, and 20%. However, a considerable value of compressive strength up to 2.7 MPa was reported in presence of 15% of any additive material with CKD-30% LPS matrix. This study recommends implementing a viable strategy to upcycle any of the examined wastes of the optimum ratios (15% waste glass or iron slag with 30% of LPS) together with another hazardous waste, namely cement kiln dust, to produce lightweight cementitious bricks in eco-friendly sustainable technology.

**Keywords:** cement kiln dust; polystyrene; iron slag; waste glass; mechanical integrity; cementitious brick

## 1. Introduction

An environmentally friendly option for disposing of industrial waste is recycling or even upcycling instead of landfilling to generate or save energy in material- and energy-intensive industries such as the cement industry in many countries. In recent years, the sustainable disposal of industrial waste has been improved by innovative economic technologies that add value by converting waste to energy [1] or reusing cementitious waste to save energy consumed for producing new virgin materials [2,3].

Mechanical parameters such as tensile strength, compressive strength, bond shear, creep, thermal expansion, and hardness are important indicators to evaluate the cementitious composite material. Furthermore, new cement-based materials need to be subjected to a variety of test conditions, including thermal cycling, gamma- and ultraviolet irradiation, biodegradation by fungi and bacteria, and internal and external chemical corrosion, all of which have to be followed by mechanical testing.

Cement is the single most used material in the construction industry. For every ton of cement produced, 0.6 to 0.7 tons of cement kiln dust (CKD) is produced; since a high volume of cement is produced, the global volume of CKD generated is 2.4 to 2.8 billion tons yearly [4].

Plastics from fossil resources constitute economical materials that have many advantages, such as being lightweight and easy to produce; however, compared to biological materials, plastics also produce enormous amounts of problematic waste due to their high

recalcitrance towards biodegradability. Therefore, the disposal or burial of this waste in garbage has led to a serious environmental problem that affects the sustainability of the planet and urgently needs to find continuous innovative solutions to overcome this threat with green and environmentally friendly approaches [5].

Plant fibers as a reinforcing material in cement-based materials is a new emerging topic to dispose of agricultural waste and to improve the mechanical property of cementitious compounds. Several studies have reported the incorporation of natural plant fibers as hazardous waste that produced through wastewater treatment [6,7], into cement concrete [8,9]. Various plant fibers, such as cellulosic fibers [10,11] or microcrystalline cellulose [12,13], are used for the generation of composites with cement.

In our laboratory, new trends have been followed in getting rid of various types of waste including problematic waste materials, such as cement kiln dust to reduce the use of cement [2,3,14] or combining nanomaterials with cement [15,16] or with polymers [17,18] for using them as stabilizing material for hazardous wastes or to produce lightweight bricks of properties suitable for construction applications. Natural materials such as clay [19,20] and bitumen [21,22], asphaltene, or poly(vinyl chloride) [23] have been blended with Portland cement to produce various composites for application in nuclear safety. Moreover, crushed waste glass was mixed with the main construction component, cement, to assess the optimum glass-to-cement ratio for the mechanical integrity of the proposed radiation shielding materials [24]. Recycled coal bottom ash [25], waste lathe scraps [26], waste lathe fibers [27], steel fibers extracted from waste tire [28] were utilized in reinforced concrete beams as replacement for aggregate.

The compressive strength, density, porosity, and water absorption of several samples of diverse compositions were measured to evaluate mechanical and physical properties. Based on national and international standards, the acquired sustainable building materials have attained the recommended value for construction applications on external shielding and internal non-load-bearing walls. Moreover, the standard values of stabilizing materials for immobilization of hazardous wastes including radioactive waste have been reached, too.

Extensive research has been conducted on the use of polymers as organic additives to improve cement compounds [29]. To reduce the porosity and fill the pores in cement-based materials used in various applications such as construction materials or hazardous waste immobilization, both unsaturated styrene polyester and polymethyl methacrylate were used [30]. For the addition of organic polymers, two distinct procedures were used, depending on their viscosity [31]: (i) impregnation process was used to introduce low-density polymethyl methacrylate, (ii) high-density styrene polyester was mixed with cement paste as a pre-mixing process.

Expanded polystyrene (EPS) comprises more than 14 million tons produced annually around the world [32]. It is considered a low-cost, high-strength thermoplastic with low thermal conductivity and used as a lightweight insulation material in a variety of applications [33].

Two problematic wastes have been utilized in construction materials, polystyrene used as a very lightweight thermal insulating material with cement [34,35] and CKD as lead-bearing material with thermally treated sewage sludges was used in light brick production [36]. However, in recent studies published by the authors, the two wastes (3% grated polystyrene waste relative to CKD) were mixed to produce an eco-friendly product of low-cost and reasonable efficiency used in building and construction applications with achieving low energy consumption and eco-sustainability [2,3].

The novelty in the current research is evident in the use of ten times the weight of the polystyrene waste used in the previous research, and then liquefied before mixing with CKD (30% liquefied polystyrene relative to CKD). A sustainable composites of various waste materials of CKD, liquefied polystyrene (LPS) waste with the addition of waste glass, cement or iron slag have been investigated to produce more durable new lightweight low-bearing bricks of proper mechanical and physical performance, at the same time achieving environmental benefit by disposing different hazardous and problematic wastes by ecofriendly technique instead of landfilling.

## 2. Materials and Experimental Approach

### 2.1. Materials

In this study, CKD, the cementitious waste used, was supplied by Torah Portland Cement Co., Egypt. This waste has low concentrations of binding compounds and so requiring binding action to increase its mechanical integrity [37]. The second waste material was PS foam, which is considered municipal waste that accumulates in problematic amounts annually as reported by US Environmental Protection Agency (EPA).

PS foam contains approximately 99% of empty voids and only about 1% of the solid material. Generally, dissolution of PS is performed by using different mixtures of acetone and toluene [17], or acetone and ethyl acetate in different proportions [38]. In this study, commercial gasoline (RON 92) was utilized as a solvent for this process to achieve economic benefit with the sustainable target.

To achieve proper mechanical integrity, commercial Portland cement (CEM 1 42.5 N) is used as a binding agent and one of the improving additives. The other two additives are iron slag and waste glass, which are mixed into the cementitious paste after being milled into fine aggregates to improve the mechanical integrity and physical properties of the produced composites.

### 2.2. Experimental Approach

#### 2.2.1. Sample Preparation

Pieces of PS waste were grated and immersed in gasoline to be liquified as a clear honey-like liquid before mixing with CKD and other improving additives to produce cementitious composite product as shown in Figure 1.

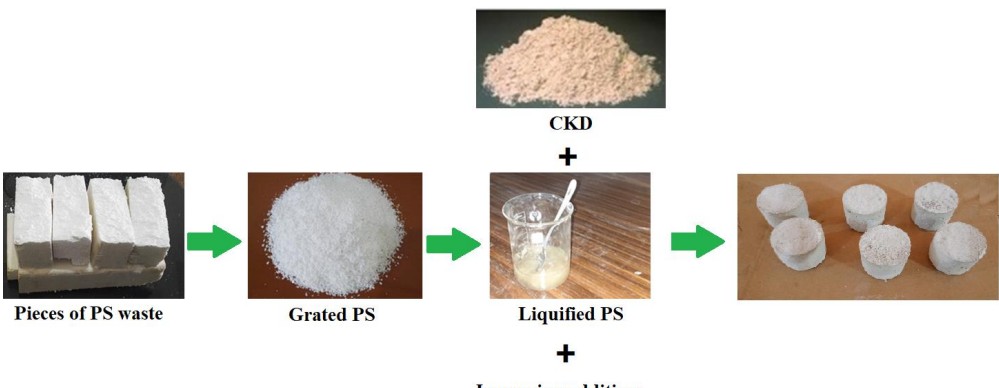

**Figure 1.** Flow chart explaining the methodology begging from the PS waste to the final product.

Degassing and liquefication of PS were performed by using gasoline (RON 92) 1:2 (wt./vol.) to prepare another series of samples as a commercial recycling process to save the spaces occupied by foams and to enable mixing higher amounts of PS waste with CKD.

In this step, the PS waste was mainly dissolved by using gasoline (RON 92); different ratios of thus liquified PS have been combined as viscous liquid with a prepared paste of cement kiln dust hydrated by water.

(90 g of cement kiln dust + 63 mL of water) + (10 g of PS + 20 mL of gasoline)

where: 70% of water = 63 mL, 2 mL of gasoline is required to dissolve 1 g of PS.

To prepare 6 samples for this test, firstly, the cement kiln dust has to be hydrated with a certain amount of water and at the same time, the predetermined amount of PS was dissolved by gasoline. The two pastes must be rapidly mixed and poured into molds before drying. The cylindrical molds (about 4 cm diameter and 6 cm height) were left closed to harden for 28 days, then stored without cover for a week to shrink and to easily remove the samples. Then, they were taken out the molds and left for a month to evaporate in the air the excess of unreacted gasoline and to obtain completely dry samples.

The other three groups of cementitious samples were prepared with hydration water of 70% relative to CKD; the optimum value of liquified polystyrene (LPS) waste and different ratios of other improving additives such as waste glass, iron slag, or Portland cement was added as reported in Table 1.

**Table 1.** Groups of CKD-LPS including different additives.

| Groups | CKD | Water | LPS | Portland Cement | Iron Slag | Waste Glass |
|--------|-----|-------|-----|-----------------|-----------|-------------|
| I | 200 g | 140 mL | 60 g | 10, 20, 30, 40 g | 0 | 0 |
| II | 200 g | 140 mL | 60 g | 0 | 10, 20, 30, 40 g | 0 |
| III | 200 g | 140 mL | 60 g | 0 | 0 | 10, 20, 30, 40 g |

2.2.2. Investigation of Compressive Strength, Porosity, and Spectroscopic Analysis

An Italian Ma-Test measuring machine E-159 SP, the model of compression apparatus, was used to apply a hydraulic press technique for measuring the compressive strength of at minimum 3 samples for each series.

By applying Archimedes saturation mechanism [39], the basic idea of the water displacement method, the mass of dry and immersed porous samples in water has been computed to evaluate the apparent porosity P, bulk density, and water absorption as reported in previous literature [40].

The small pieces of fractured samples that had undergone compressive strength tests were studied by a Philips XL 30 scanning electron microscope regarding the internal microstructure. Some fractured pieces of cementitious specimens were milled to be mixed with KBr for FTIR spectroscopic analysis using a Fourier transform infrared spectrophotometer (FTIR-8201PC, Shimadzu, Tokyo, Japan).

**3. Results and Discussion**

*3.1. Mechanical Properties Based on Compressive Strength and Porosity*

3.1.1. Blending of CKD with Various Ratios of LPS

LPS waste was mixed with CKD as filler material without any pozzolanic properties to support more binding and low voids. Various ratios of LPS waste (10–50%) were extensively mixed with a constant amount of CKD hydrated with 70% water; the water used for hydration increases with the quantity of CKD due to the presence of high amount of alkalis, sulfates, volatile salts, and lime in cement dust, as well as the high surface area leading to high water demand as explained in previous literature [41]. The mixture was poured into several plastic cups of the same geometry and volume acting as molds and cured for 28 days to achieve complete reaction and drying.

The compressive strength and porosity of CKD-LPS composites are presented in Figure 2. The compressive strength shows a gradually slightly increasing trend until reaching the maximum value of 2.6 MPa at 30% addition of LPS. Hence, 30% LPS was indicated as the appropriate amount to be mixed with CKD to provide a reasonable compressive strength with expedient thermal and sound isolation properties. Previous research has found that the thermal conductivity coefficients of cementitious samples containing PS are lower than those of other construction materials that do not contain PS [42]. With increasing the added amount of LPS, the heterogeneity of the composite material increases with a negative effect and the pores within cementitious components get filled with the liquified polymer, causing a positive effect on the mechanical integrity to attain the optimum value of compressive strength at 30% LPS. At a higher percentage of added LPS, consequently, the value of CKD has been decreased relative to LPS to reach 50%, which goes in parallel with a decrease in compressive strength.

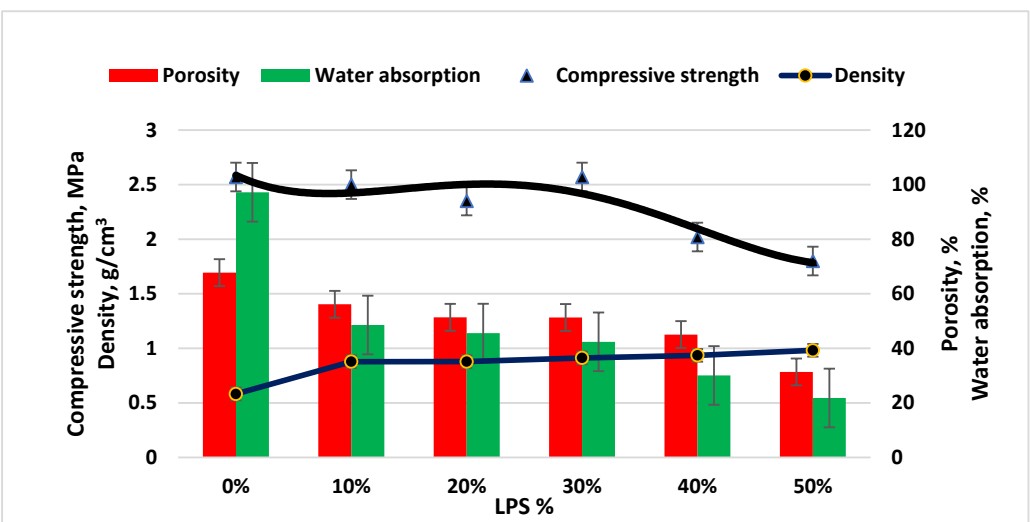

**Figure 2.** Compressive strength, porosity, density, and water absorption of the CKD-LPS composite.

On the one hand, this decrement is independent of the filling of pores present within the CKD granules that are reflected in decreasing the porosity. In agreement with recent publication, porosity decreases with more addition of liquified polystyrene to CKD, while in previous studies, the porosity increases significantly with more incorporation of grated PS with CKD [43]. Similarly, water absorption decreases with increasing the amount of added LPS while maintaining the same CKD value due to the indirect reflection of absorption affinity to the porosity due to the permeable pore volume and its connectivity as reported in previous study [44]. On the other hand, the bulk density of hardened cementitious composites has gradually increased as the amount of LPS in the mixture has increased due to the more complete filling of voids and pores within the samples.

These findings imply that another mixture is required to improve the mechanical, physical, and chemical properties of the nominated composite of the two problematic wastes (CKD and LPS) to meet the standards and recommended values of construction materials suitable for lightweight concrete and low-bearing bricks. Other waste materials, such as iron slag, waste glass, or cost-effective materials such as Portland cement, must be mixed in various amounts with the two major wastes (CKD and LPS) to compensate for the shortcomings caused by the hydrophobic nature of LPS by improving the cohesiveness of the newly produced composites.

3.1.2. Incorporation of Iron Slag into CKD-LPS Paste

With the growing need for low-cost, low-$CO_2$ concrete, there is still a lot of interest in incorporating various industrial wastes into concrete to improve performance under aggressive conditions [45]. Several waste metals have been used as an additive in the production of cement such as nickel slag [46], or to improve the integrity of cement paste such as recycled copper slag [47], replacing Portland cement and granulated blast furnace slag with CKD at a maximum value of 10% [48], or cement mortar incorporating a high volume of granulated blast furnace slag fraction and CKD (60%) [49].

In this study, CKD is the major ingredient with 30% LPS and another additive is blast furnace slag fraction with various values up to 20% without including Portland cement. During these experiments, three specimens were subjected to compressive strength testing and the other three specimens have been subjected to porosity and water absorption tests.

As shown in Figure 3, the density increases steadily with increasing the mass of iron slag to reach the maximum value of 20% iron slag. Due to the alkaline environment of cementitious hydration, the hydroxides in hydration products, such as $Ca(OH)_2$ and NaOH, are increasingly dissolved in the pore solution, increasing the alkalinity of the pore solution. This alkaline solution increases the porosity and at the same time can protect the iron slag against corrosion by forming a layer of corrosion products on the surface of

the iron [50]. Consequently, the compressive strength of the cementitious product was improved to 3.6 MPa by increasing the addition of iron slag up to 15%; then, compressive strength decreases as a result of a significant increase in porosity (69.5% at the addition of 20% iron slag). Compared with the recent previous study [2], the compressive strength did not exceed 1.50 MPa in the case of incorporating only 3% solid grated PS in CKD in the presence of 15% iron slag; this result confirms the benefits of using PS waste in liquid form instead of solid form to improve the compressive strength and durability of the produced cementitious composite.

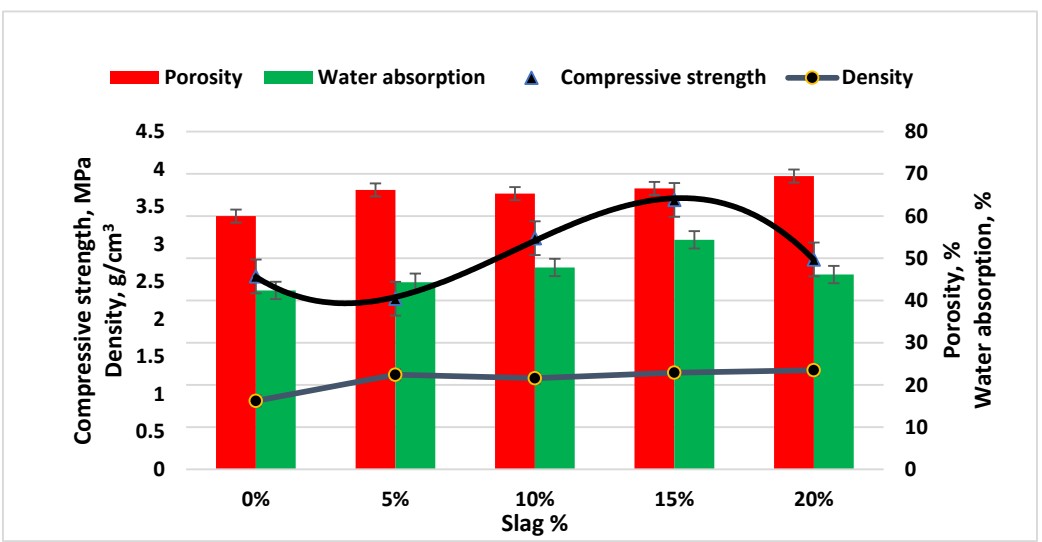

**Figure 3.** Compressive strength, porosity, density, and water absorption of the CKD-LPS in presence of iron slag.

This behavior could be explained by the enhancement of the pozzolanic reaction of the iron slag granules in the presence of the alkaline materials and sulphates produced from CKD during 28 days of curing [49]. As a result, the utilization of iron slag and CKD-LPS as a low-bearing construction material that can perform similarly to virgin cement could be a promising solution.

### 3.1.3. Incorporation of Waste Glass into CKD-LPS Paste

Cement and glass production sectors face many problems, such as large emissions of greenhouse gases, extensive use of energy, and extensive use of the earth's natural resources. On the other hand, burying the waste from these two industries of CKD cement as by-pass or glass aggregates is also an environmental threat that calls for environmentally friendly management of these non-biodegradable wastes [51]. Recycling these wastes preserves natural resources, reduces landfill space, and saves energy and money. Due to the pozzolanic activity of silica fumes and waste glass, these materials are preferred additives for improving the mechanical integrity of cement and concrete compounds [52,53], as well as a partial cement replacement in the mortar composition [54].

In the current study, waste glass was ground and mixed extensively with CKD-LPS to produce a homogeneous paste with modified mechanical properties proper for lightweight concrete constructions. Figure 4 shows the effect of different amounts of waste glass added to CKD-LPS under investigation of various mechanical parameters.

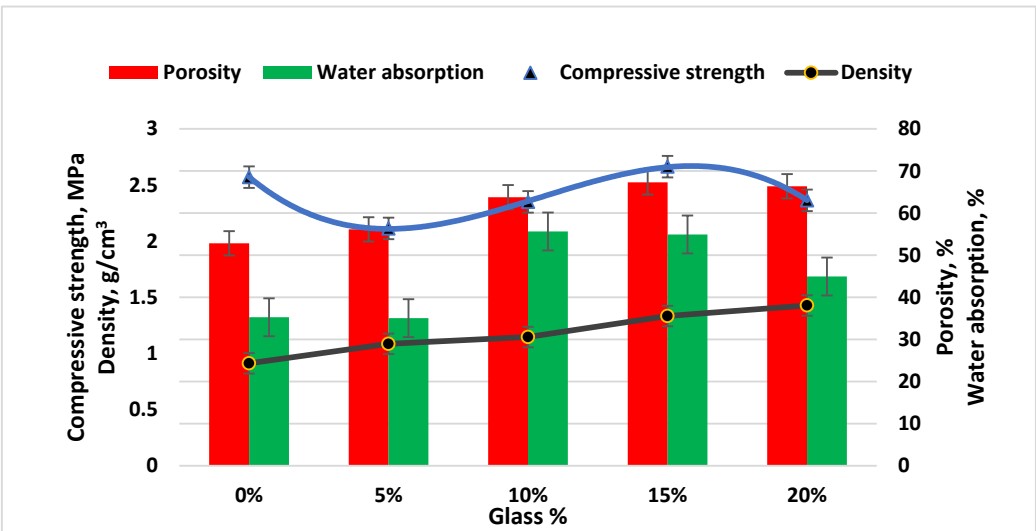

**Figure 4.** Compressive strength, porosity, density, and water absorption of the CKD-LPS in presence of waste glass.

A gradual increase in the density of the cementitious samples has been detected with increasing the added value of waste glass due to the higher density of glass relative to CKD or LPS. The compressive strength, which is the most essential indicator of mechanical integrity, exhibited only a slight increase, reaching 2.66 MPa when mixing 15% waste glass with the CKD-LPS. Although this value is not very high, it is higher than the compressive strength (1.6 MPa) when using 3% grated PS in solid form under the same conditions [2]. According to the previous interpretation in the case of grated PS, the porosity and water absorption decreased with an increase in the value of the ground glass due to the possibility of filling holes and pores inside the cementitious paste [2]. In the case of liquefied PS, with increasing amounts of mixed waste glass, a steady increase in porosity and water absorption can be observed due to the inhomogeneity between the two phases.

As a result, the existence of ground waste glass in this composition can be viewed as an additional benefit of combining another waste material in addition to CKD-LPS; however, another material must be added to improve the mechanical property, e.g., Portland cement, which can result in appropriate mechanical properties and achieve more ecological and environmental benefits.

### 3.1.4. Improvement of CKD-LPS Paste by Addition of Various Amounts of Portland Cement

To reach the target of this study of improving the mechanical property of the composite material produced from two types of wastes, Portland cement was additionally added. Figure 5 shows various parameters of the cementitious samples with different ratios of Portland cement added to the original paste. As expected, density increases as the content of Portland cement increases. The composite of 20% volume Portland cement had a density of 1.46 g/cm$^3$ compared to 0.9 g/cm$^3$ without the addition of Portland cement. This means that cement has a significant influence on the density of samples. In this direction, the compressive strength was increased gradually by increasing the cement addition to reaching the maximum value of 2.71 MPa at an additive (Portland cement) ratio of 15%. In agreement with this behavior, Al-harthy et al. reported that concrete mixtures including cement with variable ratios of CKD (5–30%) have lower strength properties and absorption characteristics with decreasing the ratio of cement relative to CKD [55]. Portland cement was included to improve the mineral binding and to increase the ability to improve the hydraulic integrity of the cementitious product.

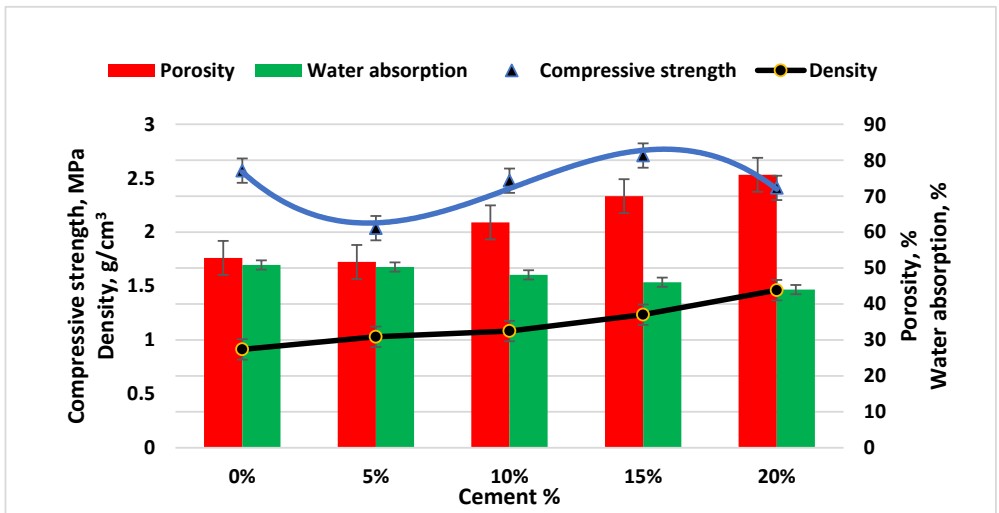

**Figure 5.** Compressive strength, porosity, density, and water absorption of the CKD-LPS in presence of Portland cement.

The compressive strength of composite CKD increased with cement as a partial replacement due to the decrease formation of sulfoaluminate, chloroaluminate as well as carboaluminate hydrates in addition to the increase of C-S-H binding centers in the cementitious pastes. This improvement in compressive strength is proportional to the cement content, which affects the mechanical properties of the hardened paste. These results are in good agreement with those reported by Khalil et al. [48].

By increasing the addition of Portland cement with CKD-LPS paste, a gradual increase in porosity was detected with a detectable decrease in the water absorption. However, as evidenced by the relative increase in compressive strength, Portland cement has improved the mechanical properties of the proposed mixture of two waste materials; thus, an appropriate portion of Portland cement (15%) can be introduced into the CKD-LPS paste to achieve the required level of mechanical integrity. The hydration reactions caused by the structural development of Portland cement-based CKD-LPS resulted in stiffening and densification as a monolith, imparting reasonable structural integrity to the composite material [37].

*3.2. FTIR Analysis CKD-LPS with Various Additional Materials*

FTIR analysis was performed to investigate the functional groups of hydrated cement, hydrated CKD, CKD-30% LPS, CKD-30% LPS + 10% cement, CKD-30% LPS + 10% iron slag, and CKD-30% LPS +10% waste glass as shown in Figure 6.

The FTIR absorption spectrum of the hydrated cementitious compounds has a modest sharp peak near 3642 cm$^{-1}$, which may be due to the formation of an OH bond during the hydration process. The stretching vibration peak of at 3425 cm is probably related to the surface OH participation of hydrogen bonds as explained in previous study [56]. Portlandite (Ca(OH)$_2$), which is formed simultaneously with the setting and hardening of cementitious components, serves as a foundation for the subsequent development of calcite (CaCO$_3$), which is carbonated during the absorption of atmospheric carbon dioxide. Calcite is distinguished by a prominent sharp absorption band at about 1420 cm$^{-1}$ and a sharp peak at 875 cm$^{-1}$. The absorption band at 966 cm$^{-1}$ of SiO could indicate the development of CSH gel during the cement-water reaction. The SO$_4$ group absorption band at about 1115 cm$^{-1}$ characterizes the SO bond in the sulphate of ettringite. In the hard-cured cement sample, an absorption band assigned to silicates can be seen near 1000 cm$^{-1}$.

The peaks assignments were nearly identical in all of the examined specimens, indicating that integrating LPS within cementitious compounds has no negative effects. The polymer basic absorption peaks have been discovered in the spectra, even though the distinctive bands for polymer moiety in the matrix interfere with bands of the phases in hydrated cement. The following are the peaks of interest that characterize the obtained polymer:

The occurrence of aromatic C-H stretching vibration is clearly indicated by the peak near 3058 cm$^{-1}$. The stretching vibration of the methylene (CH) groups is responsible for the bands about 2925 and 2850 cm$^{-1}$. The aromatic C=C stretching absorption is confirmed by the peak at 3025 cm$^{-1}$. The PS polymer is primarily responsible for the peak of about 1600 cm$^{-1}$, while the monosubstituted benzene is characterized by a band near 699 cm$^{-1}$.

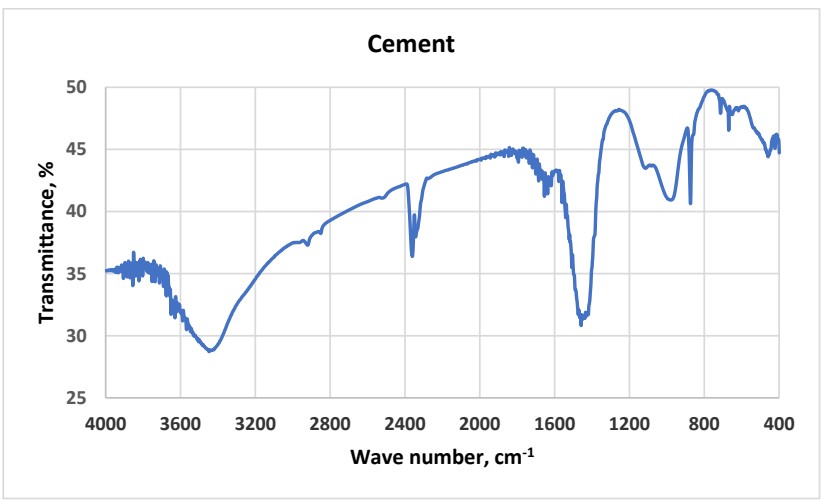

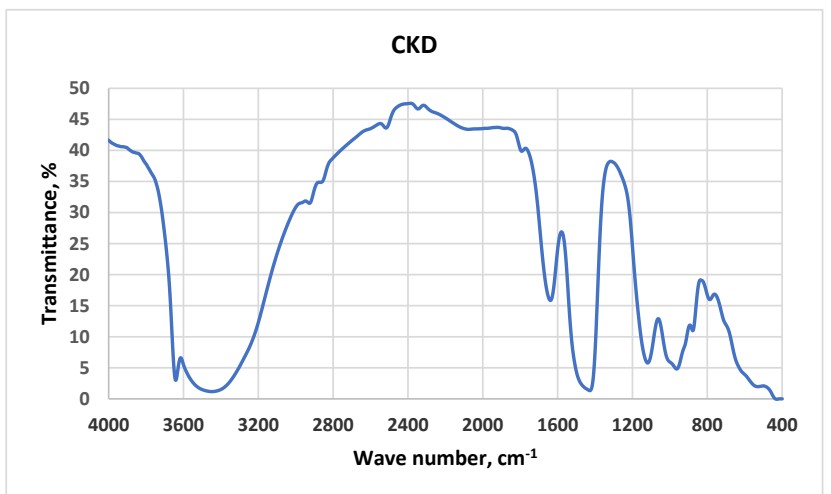

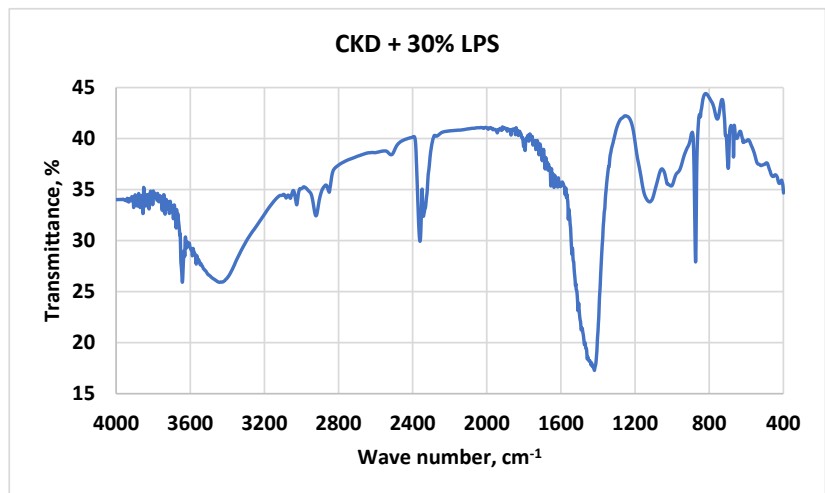

**Figure 6.** *Cont.*

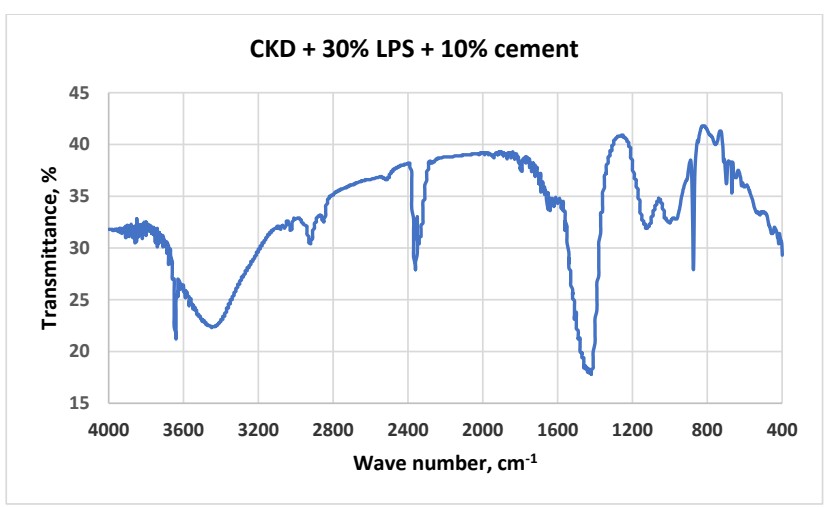

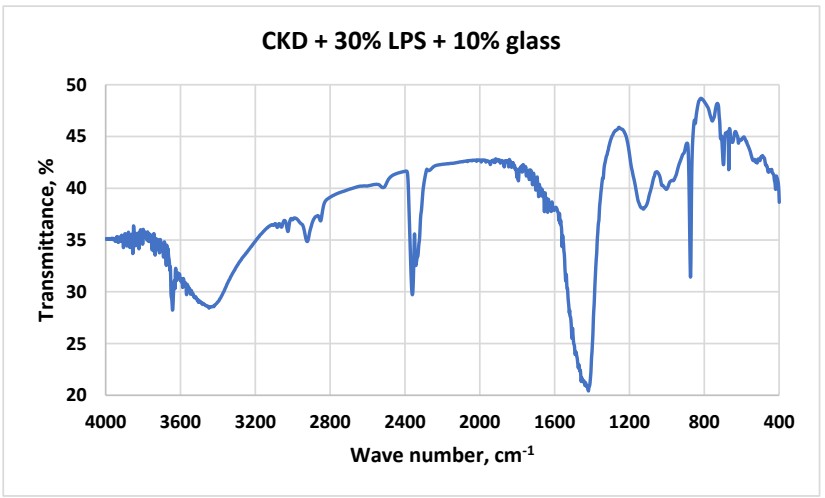

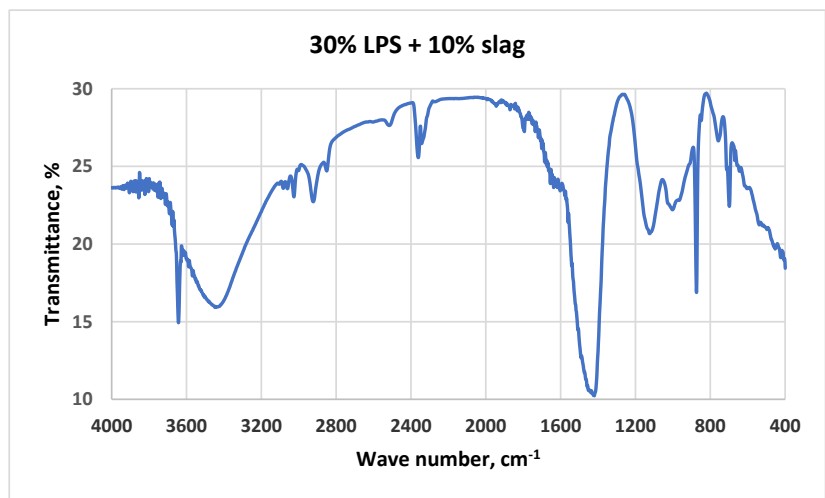

**Figure 6.** FTIR of various cementitious compositions including the CKD-LPS composite.

According to this FTIR investigation, the dissolution of the PS waste by gasoline (RON 92) achieved a volume reduction of more than 99 percent without any degradation in the characterization of the polymer chain.

### 3.3. Morphological Characterization by SEM Investigation

The SEM examination of the cementitious specimens revealed that the internal structure was quite homogeneous even after the addition of the LPS polymer and its interactions with the CKD hydration products.

The LPS paste is embedded in the cementitious matrix and distributed uniformly, with no evidence of aggregation. Furthermore, as shown by the SEM images in Figure 7, the specimens have a compact structure with no huge cracking. This indicated that the LPS paste adhered very well to the cementitious matrix and consequently the polymeric paste acts as an adhesive that accumulated the gradients of the cementitious compound with the other additive materials; the mechanical properties are improved by the strong adhesion as described in previous study [57].

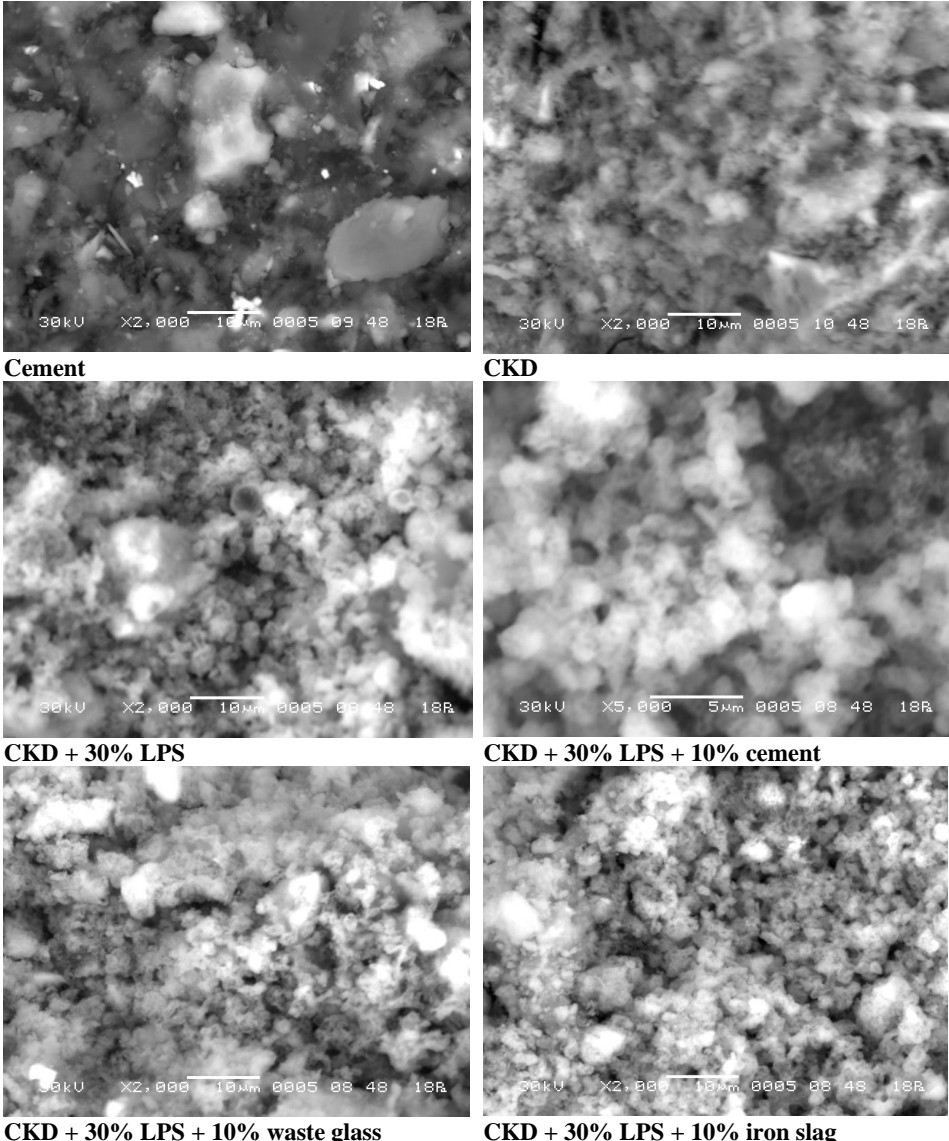

**Figure 7.** SEM of cement, CKD, and composites of CKD-LPS in the presence and absence of other additive materials.

In other words, the polymer acts as a binding agent for various hydrated cementitious phases such as portlandite and calcium silicate hydrate. Furthermore, ettringite and some salt crystals appear to deposit around the composite pores. These processes explain and confirm the improvement in mechanical parameters as well as the decrease in water absorption capacity in the presence of LPS.

## 4. Conclusions

The primary goal of this study was to develop an eco-friendly, low-cost, lightweight concrete-based composite material for utilization as external thermal isolation and in internal low-bear walls. A novel lightweight sound and thermal insulating material was prepared and characterized in the presence and absence of different additive materials, based on polymer-concrete composites containing liquified polystyrene (LPS) as insulating aggregate and cement kiln dust (CKD) as a byproduct generated by the cement industry.

The major conclusions could be summarized as:

1. The microstructural characterization showed a homogeneous structure with LPS uniformly dispersed and embedded in the cementitious matrix.
2. Compressive strengths increased gradually to approach the maximum value at 30% LPS content and then decreased with more addition of LPS.
3. The addition of Portland cement and other industrial waste such as waste glass and iron glass to the CKD-30% LPS has significantly increased the compressive strength, reaching the optimum value at 15% of additive material and then followed by sudden decrease at 20% of addition.
4. The addition of 15% of any additive material had an appropriate distribution in the CKD-30% LPS matrix with a significant value of compressive strength up to 2.7 MPa.

This research has demonstrated the viability of the production of lightweight cementitious composites of sound and thermally insulating property with suitable mechanical integrity.

Furthermore, the environmental and economic benefits suggest that CKD-LPS lightweight concrete can be used as a commercial ecological material in the production of bricks for internal low-bear constructions. The novelty in this study is in conducting a feasible approach for the incorporation of various waste materials within the CKD-LPS pastes to achieve even more mechanical integrity instead of landfilling these wastes.

**Author Contributions:** Conceptualization, H.M.S.; Methodology, A.M.E.-S. and H.M.S.; Writing—original draft, H.M.S.; Writing—review & editing, H.M.S.; Supervision, A.A.F., A.A.S. and H.M.S. All authors have read and agreed to the published version of the manuscript.

**Funding:** This research received no external funding.

**Conflicts of Interest:** The authors declare no conflict of interest.

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
