# Peer review of "Sustainable Lightweight Concrete Made of Cement Kiln Dust and Liquefied Polystyrene Foam Improved with Other Waste Additives"

_sustainability, doi:10.3390/su142215313_

Round 1
Reviewer 1 Report
Please find my comments in the file

Author Response
Dear reviewer,
Thank you for your time and your useful comments that certainly will improve my submission.
All suggested modifications were presented in the attached file and reflected in the revised submission.

Reviewer 2 Report
Lines 77-84: Indicative references should be added.
Lines 89-95: This section should be further enhanced with more references. Also, what is the research gap the proposed research attempts to fill? What is its connection to previous literature in terms of research objectives?
Lines 106-107: This sentence needs rephrasing.
Lines 123-125: This sentence needs rephrasing.
Line 127: Please capitalize the first letter of the first word.
Lines 170, 291: There are too many spacings.
Lines 214-237: Why is this text bold and aligned to the left?
Line 246: "wastes"
Line 355: "additive materials"
Author Response
Dear reviewer,
Thank you for your time and your useful comments that certainly will improve my submission.
All suggested modifications were considered.

Reviewer 3 Report
The authors conducted experimental study using waste materials for lightweight concrete however there are major issue must be solved before publication.
Abstract should contain important results not general ones.
Novelty is not clear in the introduction. It is stated there are limited studies. What is the difference between this and other studies.
What are the size of samples
What is workability and slump test?
Add actual photo for test setup
Add actaul photo of samples
Add damaged photos of samples
The reason for selectting Mix design should be added in detail. Ratio of cement, aggerate water and etc.
The importance of recycling materials to overcome enviermental prbolem should be added to introduction using: influence of replacing cement with waste glass on mechanical properties of concrete; use of recycled coal bottom ash in reinforced concrete beams as replacement for aggregate; improvement in bending performance of reinforced concrete beams produced with waste lathe scraps; performance assessment of fiber-reinforced concrete produced with waste lathe fibers; performance evaluation of fiber-reinforced concretes produced with steel fibers extracted from waste tire; performance evaluation of fiber-reinforced concretes produced with steel fibers extracted from waste tire
Add recent studies on this subject to introduction. There are many studies on the introduction for this topic.
Conclusion should be improved. The recommendation consdiering all test should be given for engineers.
Author Response

(The authors gave the same response as above.)

Round 2
Reviewer 3 Report
The paper can be accepted in this current form.